# Use of Robot-Assisted Ankle Training in a Patient with an Incomplete Spinal Cord Injury: A Case Report

**DOI:** 10.3390/jfmk8010031

**Published:** 2023-02-27

**Authors:** Kazunori Koseki, Kazushi Takahashi, Satoshi Yamamoto, Kenichi Yoshikawa, Atsushi Abe, Hirotaka Mutsuzaki

**Affiliations:** 1Department of Physical Therapy, Ibaraki Prefectural University of Health Sciences Hospital, 4733 Ami, Inashiki-gun, Ibaraki 300-0331, Japan; 2Department of Physical Therapy, Ibaraki Prefectural University of Health Sciences, 4669-2 Ami, Inashiki-gun, Ibaraki 300-0394, Japan; 3Department of Orthopaedic Surgery, Ibaraki Prefectural University of Health Sciences Hospital, 4733 Ami, Inashiki-gun, Ibaraki 300-0331, Japan

**Keywords:** incomplete spinal cord injury, robot-assisted training, robotics

## Abstract

Rehabilitation interventions are crucial in promoting neuroplasticity after spinal cord injury (SCI). We provided rehabilitation with a single-joint hybrid assistive limb (HAL-SJ) ankle joint unit (HAL-T) in a patient with incomplete SCI. The patient had incomplete paraplegia and SCI (neurological injury height: L1, ASIA Impairment Scale: C, ASIA motor score (R/L) L4:0/0, S1:1/0) following a rupture fracture of the first lumbar vertebra. The HAL-T consisted of a combination of ankle plantar dorsiflexion exercises in the sitting position, knee flexion, and extension exercises in the standing position, and stepping exercises in the standing position with HAL assistance. The plantar dorsiflexion angles of the left and right ankle joints and electromyograms of the tibialis anterior and gastrocnemius muscles were measured and compared using a three-dimensional motion analyzer and surface electromyography before and after HAL-T intervention. Phasic electromyographic activity was developed in the left tibialis anterior muscle during plantar dorsiflexion of the ankle joint after the intervention. No changes were observed in the left and right ankle joint angles. We experienced a case in which intervention using HAL-SJ induced muscle potentials in a patient with a spinal cord injury who was unable to perform voluntary ankle movements due to severe motor–sensory dysfunction.

## 1. Introduction

Traumatic spinal cord injury (SCI) is caused by physical impact on the spinal cord due to several reasons, such as road traffic trauma and falls from heights. In the United States, the estimated incidence rate of SCI is 52–54 cases per million people [1,2], while in Japan, it is approximately 49 cases per million people [3]. The primary pathology of SCI is damage to the spinal cord due to primary trauma and subsequent secondary injury, which results in the blockage of descending and ascending nerve conduction from the brain, resulting in motor and sensory deficits, bladder–rectal deficits, and autonomic neuropathy, following the injury [4]. This causes dysfunction in the ankle joint, which is dominated by the lumbar and sacral spinal cord regions, one of the most vulnerable areas to impairment. Physical and occupational therapies, including mobility exercises using a wheelchair and activities of daily living exercises, are commonly used in patients with paraplegia due to SCI in the lumbar region [5]. In addition, clutches and lower limb orthoses are generally used to compensate for missing ankle joint function during standing and walking practice [5]. In recent years, because of advances in robotics, the effectiveness of interventions using various exoskeletal robots has been verified [6,7]. However, most of these assist with walking movements, and there have been few reports on robotic interventions specifically designed to improve ankle joint function.

The hybrid assistive limb (HAL) has a bioelectrical signal (BES)-based control system that can be assisted by a joint torque driven voluntarily by the wearer. The single-joint-type HAL (HAL-SJ, Cyberdyne, Inc., Tsukuba, Japan) is a robot that can support flexion and extension movements of various joints. Previously, interventions for elbow and knee joint dysfunction have been reported [8,9,10,11], but with the expansion of the ankle joint unit, it is being explored for ankle joint dysfunction [12]. Furthermore, the use of HAL-SJ has been reported to improve the function of elbow joint muscles that failed to contract voluntarily in patients with cervical cord injury (C4 level) [13]. Thus, repetition of voluntary exercise using HAL-SJ can improve paralyzed muscles in patients with spinal cord injuries. Hence, this study aimed to perform an intervention (HAL-T) using a single-joint HAL ankle joint unit in a patient with paraplegia due to SCI.

## 2. Case Report

### 2.1. Participant

The study patient was a 34-year-old man (height, 169 cm; weight, 79.4 kg). He only had pre-existing medical history of hyperlipidemia. He sustained a burst fracture of the first lumbar vertebra, dislocation of the right shoulder joint due to a fall from a height. At the time of emergency transport, the patient was found to have bilateral lower extremity paralysis and cysto-rectal disturbance, and was diagnosed as having a spinal cord injury. As assessed by The International Standards for Neurological Classification of Spinal Cord Injury, the neurological level of injury at the time of injury was the second lumbar (L1) level, and the American Spinal Cord Injury Association (ASIA) Impairment Scale was B (sensory but not motor function is preserved below the neurological level and includes the sacral segments S4–S5, and no motor function is preserved at more than three levels below the motor level on either side of the body) [14]. The patient was transferred to an acute-care hospital and underwent posterior spinal decompression fusion on the same day (Figure 1a,b). Then, at 18 days after the injury, he was transferred to our hospital for continued rehabilitation. Physical and occupational therapy was initiated at the hospital, focusing on muscle training, standing exercises, walking exercise with orthosis, and activities of daily living (ADL) exercises. On the 101st day post-injury, the patient’s neurological injury level was L1, the ASIA Impairment Scale was C, and the ASIA motor score (R/L) was as follows: L2: 4/4, L3: 3/4; L4: 0/0; L5: 0/0; and S1: 1/0; the patient had symptoms of paraplegia. The patient was able to perform basic ADL in the hospital using a wheelchair and was able to walk with a walker using a knee–ankle foot orthosis on both the right and left sides. However, the patient’s ankle joint function remained severely impaired, and the patient was highly concerned about recovering the function. Therefore, an intervention using a single-joint HAL ankle unit was performed, in addition to the aforementioned usual rehabilitation programs to improve the ankle joint function.

### 2.2. Intervention Used

HAL-SJ allows the wearer to voluntarily perform active assisted exercises through BES-based control. The HAL-SJ electrodes were affixed to the tibialis anterior (TA) and the gastrocnemius (GAS) muscles and driven based on their BES. HAL-SJ was applied to both ankle joints. The HAL-SJ intervention consisted of a combination of (1) ankle joint plantar dorsiflexion exercises in the end-sitting position (Figure 2a), (2) squat exercise, and (3) stepping exercises (Figure 2b) under HAL assistance, 20 min per session, five times a week, 10 times in total. In each intervention, the patient first performed ankle plantar dorsiflexion exercises in the end-sitting position to activate foot muscle activity and movement, followed by squatting and stepping tasks in the standing position. During squat and stepping exercises, a harness was used for fall prevention and partial unloading. Sensitivity adjustment of the amount of assist torque according to the degree of patient BES can be controlled by operating the controller and can be increased or decreased by the control item “assist gain.” To counteract undesirable assist due to abnormal BES caused by the antagonist muscles when using HAL-SJ, a control mechanism called “assist balance” can be used to reduce the BES in 20 steps from 0% to 100% in 5% increments. This control item makes it possible to adjust the balance between flexion and extension assist torques. In this intervention, the assist gain and balance were adjusted to provide the desired exercise according to the level of spasticity and BES. The assist gain and balance were changed on a case-by-case basis to allow HAL-SJ to effectively perform joint exercises.

### 2.3. Outcome Measurement

Lower limb kinematic parameters were measured, and electromyography (EMG) was performed before and after the HAL-SJ intervention period. The voluntary ankle joint plantar-dorsal flexion movements in the end-sitting position (after 10 s of rest, 10 times each of alternating plantar and dorsiflexion movements at 1 Hz timing, signaled using a metronome) were measured. Both ankle kinematic parameters and EMG were measured using a wireless inertial measurement unit (IMU) system (myoMOTION, Noraxon USA Inc., Scottsdale, AZ, USA) consisting of an IMU, and an Ultium-EMG sensor system (Noraxon Inc.) with a sampling frequency of 2000 Hz, and a bandpass filter of 10–450 Hz. IMUs were placed on the anterior surfaces of the tibia and metatarsals bilaterally to measure ankle joint angles. Each IMU had a local coordinate system and measured acceleration. The joint angles between the IMUs were calculated by the IMU system software (myo RESERCH 3.16.86, Noraxon USA Inc., Scottsdale, AZ, USA). Lower limb kinematic parameters were resampled to 100 points of joint motion from plantar flexion to dorsiflexion and plantar flexion and averaged 10 times. EMG patterns of the bilateral TA and GAS muscles were recorded, and the raw waveform of the EMG was depicted. Then, the EMG data were rectified, smoothed (RMS algorithm with a smoothing window width of 100 ms), added, and averaged for 10 trials.

### 2.4. Outcome of the Intervention

A total of 10 sessions was performed, and no adverse events, such as abrasions, excessive muscle fatigue, or pain, were observed. In the first session, effective joint movements in the intended direction did not occur sufficiently in the end-sitting lower-limb movements. However, after the second session, joint movements under HAL-assist gradually appeared after providing positive feedback when the intended movements occurred, by visually checking the EMG graph on the HAL controller, adjusting “assist gain” and “assist balance”.

Figure 3, Figure 4 and Figure 5 show the results of lower-limb kinematic parameters and EMG during voluntary ankle joint plantar-dorsal flexion movement in the end-sitting position. Lower limb kinematic parameters showed that there was no obvious phasic joint angle change according to ankle joint plantar dorsiflexion timing on either side before or after the HAL-SJ intervention (Figure 3). Raw EMG data during end-sitting ankle plantar dorsiflexion movements showed no obvious muscle activity in the TA, bilaterally, before the intervention; however, after the intervention, muscle activity was observed in the left TA (Figure 4). Furthermore, when averaged over 10 plantar dorsiflexion exercises, phasic muscle activity was clearly observed in the left TA during dorsiflexion exercises (Figure 5). No clear muscle activity was observed on either side of the GAS after the intervention.

## 3. Discussion

The patient had suffered a lumbar fracture and spinal cord injury due to a fall from a height, and had made progress in acquiring ADLs using a wheelchair through rehabilitation. He was just beginning to practice walking with a lower limb orthosis. However, the ankle joint function had not improved sufficiently, and the patient desired further intervention. For this reason, we attempted to provide intervention using the HAL-SJ, an exoskeleton-type device driven by BES. A total of 10 HAL-SJ intervention sessions were performed, and no adverse events, such as abrasions, excessive muscle fatigue, or pain, were observed. Comparison of EMG before and after intervention revealed phasic muscle activity in the left tibialis anterior muscle. However, the current interventions did not produce joint angle changes during ankle joint motion, and did not reach the point where changes in walking ability or ADL occurred.

In this case, no obvious joint movement was observed in the cybernic voluntary control mode driven by the BES at the start of the intervention. However, by adjusting the assist gain, assist level, and assist balance of the HAL to create a state where the joint movement was likely to occur, by visually presenting the EMG graph on the controller and providing positive feedback when the intended joint movement occurred, the voluntary movement gradually appeared. Training duration, high intensity, and augmented feedback are among the factors that have been reported to influence training effectiveness after SCI [15]. In addition to electrical stimulation and biofeedback therapy [16,17,18,19], the effectiveness of various types of robotic training [7] has also been reported in patients with SCI and sensorimotor impairment. The concept of plasticity-based functional training emphasizes that more intensive training with adjusted difficulty can be performed using a rehabilitation robot [20,21] and that robotic support should be minimized to challenge the patient’s efforts [10,22]. In the present case, the patient had severely impaired ankle sensory–motor function, and without HAL-SJ there was no clear ankle motion. Even though the patient did not sense movements, it was possible to use the equipment to generate foot motion and perform repetitive training at an optimal load with HAL-SJ. The intervention with the HAL-SJ contributed to obtaining visual and sensory feedback, the ability to exercise under optimal effort, and repeated intervention using rich feedback was effective in activating ankle joint function. In fact, before training with HAL-SJ, as shown in Figure 4, no obvious muscle activity was observed in either the left or right TA during ankle plantar dorsiflexion movement; however, after HAL-SJ intervention, muscle activity was observed in the left TA. The results of the 10-trial average showed phasic muscle activity during dorsiflexion in the left TA, suggesting an effect of training. It is not clear why muscle activity appeared only in the left TA despite interventions in both lower extremities, and the results of this study do not provide a complete explanation. In a report by Shimizu et al., HAL-SJ successfully induced the movement of paralyzed target muscles in a patient with cervical cord injury through performing of repetitive movements triggered by voluntary movements of other muscles [13]. However, in the present case, as the HAL-SJ was able to drive by the BES of the TA relatively early in the intervention; we speculate that the left lower extremity in this case had a potential for muscle contraction that was masked by the severe motor–sensory deficit. We believe that the advantage of using this device is that rich visual and motor feedback was provided to such masked potentials.

This study had some limitations. In this case, the HAL-SJ intervention was performed at approximately 100 days after injury. As this is a period of spontaneous recovery due to neuroplasticity after SCI [23], it may be difficult to completely determine the effect of HAL-SJ intervention. Therefore, it is not clear whether the effect of EMG appearance in the left TA obtained in this study was attributed to neuroplasticity or the activation and manifestation of voluntary movements that were difficult to perform due to severe motor sensory impairment. However, even after considering this issue, the appearance of EMG in the trained area was significant, and further improvements could have been achieved with prolonged training.

It should be noted that the device is subject to rental fees, which could be a potential cost and may make it difficult to use in all hospitals and institutions. Although the effects obtained with HAL-SJ alone were partial in this case, they may lead to a step up to another interventions using muscle activity. Thus, this device may be effective as one of the interventions that can be selected based on the patient’s condition. In addition, ankle joint function is closely related to standing balance [24], and improvement of ankle joint function can lead to more practical gait by improving balance during walking [25], and may have a spillover effect on ADL, such as allowing selection of a simpler lower limb orthosis in the future. In the future, the number and frequency of effective interventions should be examined. Moreover, the extent to which these interventions are effective should be further verified.

## 4. Conclusions

We experienced and described a case in which intervention using HAL-SJ induced muscle potentials in a patient with a spinal cord injury who was unable to perform voluntary ankle movements due to severe motor–sensory dysfunction. Although intervention using the HAL-SJ could contribute to inducing muscle activity in this patient after SCI, further validation is needed to prove that HAL-SJ intervention induces locomotion in patients after SCI.

## Figures and Tables

**Figure 1 jfmk-08-00031-f001:**
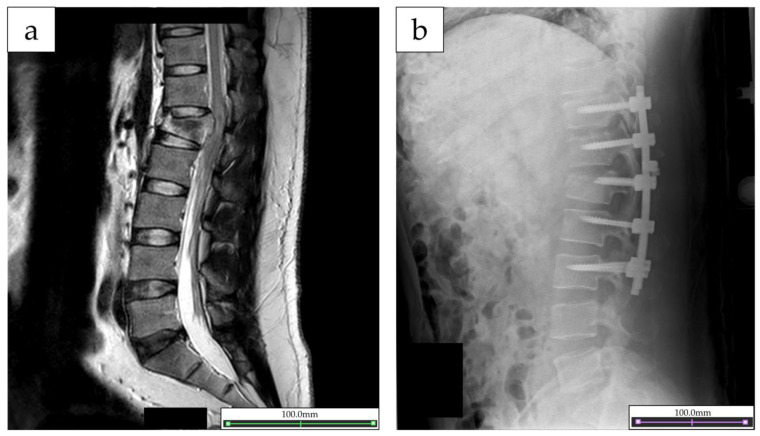
(**a**) Magnetic resonance imaging images of the lumbar spine and spinal cord taken on the date of injury. (**b**) X-ray image after posterior spinal decompression fusion.

**Figure 2 jfmk-08-00031-f002:**
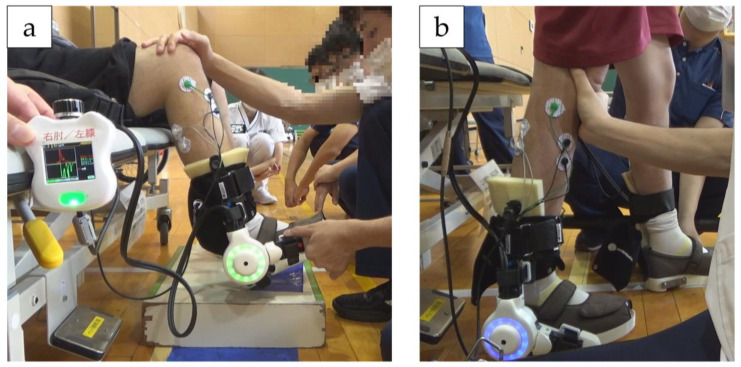
(**a**) Ankle joint plantar dorsiflexion exercises with HAL-SJ in end-sitting position. (**b**) Stepping exercises with HAL-SJ. During squat and stepping exercises, a harness was used to prevent falls and partial unloading.

**Figure 3 jfmk-08-00031-f003:**
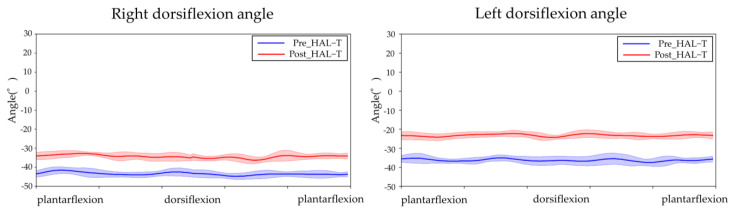
Ankle joint angle during ankle joint plantar dorsiflexion movement in end-sitting position. The blue line shows angle before HAL-T and the red line shows after HAL-T. The 10 ankle plantar dorsiflexion movements were averaged, and the results are shown as means ± standard deviations.

**Figure 4 jfmk-08-00031-f004:**
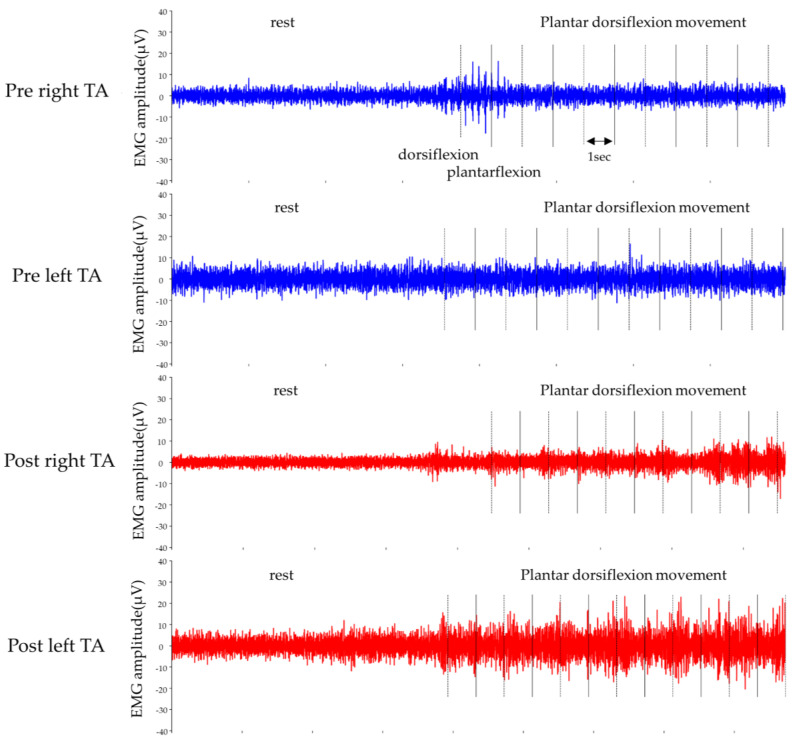
Raw EMGs of the left and right tibialis anterior (TA) muscles during ankle joint plantar-dorsiflexion movement in the end-sitting position.

**Figure 5 jfmk-08-00031-f005:**
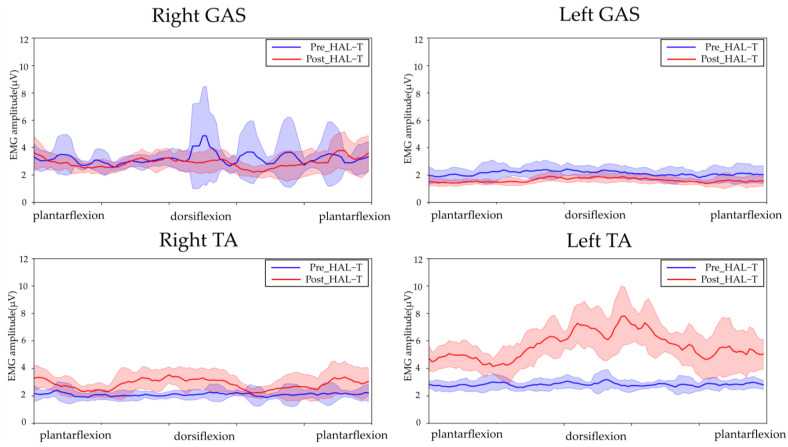
EMGs of the left and right tibialis anterior (TA) and gastrocnemius (GAS) muscles during ankle joint plantar dorsiflexion movement in the end-sitting position. The blue line shows EMG before HAL-T and the red line shows after HAL-T. The raw EMGs were rectified and smoothed (RMS algorithm with a smoothing window width of 100 ms), and the 10 ankle plantar dorsiflexion movements were averaged. The results are presented in the graph as means ± standard deviations.

## Data Availability

The data are contained within the article.

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
