# Peer review of "Use of Robot-Assisted Ankle Training in a Patient with an Incomplete Spinal Cord Injury: A Case Report"

_jfmk, 2023, doi:10.3390/jfmk8010031_

Round 1

Reviewer 1 Report

This paper reports a case report that aimed to assess the feasibility of rehabilitation with a single-joint Hybrid Assistive Limb (HAL-SJ) ankle joint unit (HAL-T) in a patient with incomplete SCI. Despite the interesting topic, the present study has a great weakness. The manuscript is very rough, and the language and writing are truly not professional so it is very hard to read. Moreover, the quality of figures is very low, and many critical data are not provided. Considering the high requirement of the Journal of Functional Morphology and Kinesiology, I would suggest this manuscript be published in a more specialized journal.

Author Response

Point 1:

 The manuscript is very rough, and the language and writing are truly not professional so it is very hard to read.

Response 1:

We would like to thank the reviewer for evaluating our manuscript and for his/her comment. We have sent our manuscript to an English editing company (Editage) for English proofreading. We hope that the level of English has been significantly improved in the revised manuscript.

Point 2:

Moreover, the quality of figures is very low, and many critical data are not provided.

Response 2:

We would like to thank the reviewer for the comment. We have verified that all images meet the minimum size specified by the journal (at least 1000 pixels wide/height or 300 dpi resolution). We also apologize for the cluttered background of some panels in Figure 2. We have cropped and reworked them. Moreover, we have provided more information concerning the participant’s age, height, weight, and medical history as well as MRI and radiographic findings to illustrate the participant’s physical condition after the injury.

Reviewer 2 Report

The current manuscript assess using of a Robot-Assisted Ankle Training in a Patient with

In complete Spinal Cord Injury Through A Case Report. Please find the following comments regarding the manuscript:

1- The case report does not assess the feasibility of an intervention. 

2- How can be performed the flexion and extension exercises in the standing position

3- what can the current manuscript add to the literature 

Author Response

Point 1:

 The case report does not assess the feasibility of an intervention.

Response 1:

The authors would like to thank the reviewer for his/her constructive critique to improve the manuscript. We have made every effort to address the issues raised and to respond to all comments. Please, find next a detailed, point-by-point response to the reviewer's comments. We hope that our revisions will meet the reviewer’s expectations.

We understood that it is difficult to assess the feasibility of an intervention in a case study. We have removed the language regarding feasibility and added a description of the circumstances that led to this patient's decision to try HAL-SJ.

Point 2:

How can be performed the flexion and extension exercises in the standing position

Response 2

As described in the "Intervention" section, the patient was placed in a standing position while wearing HAL-Sj, and squatting and stepping movements were performed. The main muscles involved in these movements are the hip and knee joint muscles. However, we considered that the activity of the ankle joint muscles is also important to control balance when standing up and stepping forward and to lift the toes when moving the lower limbs forward. Therefore, we used HAL-Sj to assist the activity of the ankle joint muscles. In addition, a harness was used to prevent falls and ensure safety.

Point 3: what can the current manuscript add to the literature?

Response 3

We would like to thank the reviewer for the question. The research result is that we are able to report that the use of exoskeletal devices driven by the EMG based system can activate the motor function of the ankle joint in a patient with spinal cord injury.

Reviewer 3 Report

In summary, this case report summarizes the use of a robot-assisted ankle training in a patient with incomplete SCI. While the manuscript itself is well written overall, I am not convinced that the case itself is interesting. Also, as the authors noted there is a possibility of spontaneous recovery. Please see some minor comments below.

Line 57: please specify age. Also, please provide more information regarding patients such as height and weight. Any specific MRI findings associated with injury?

Line 75-76:  what were usual rehabilitation programs?

Discussion: why only left TA? wasn't intervention done in both ankles? Would you be able to explain this?

Also, training with the device seems to require intensive sessions with therapists. Time and cost wise, would it this be feasible for other patients? If not, please discuss this as limitations.

Author Response

Point 1:

 Line 57: please specify age. Also, please provide more information regarding patients such as height and weight. Any specific MRI findings associated with injury?

Response 1:

The authors would like to thank the reviewer for his/her constructive critique to improve the manuscript. We have made every effort to address the issues raised and to respond to all comments. Please, find next a detailed, point-by-point response to the reviewer's comments. We hope that our revisions will meet the reviewer’s expectations.

We have provided more information concerning the participant’s age, height, weight, and medical history as well as MRI and radiographic findings to illustrate the participant’s physical condition after the injury.

Point 2:

 Line 75-76:  what were usual rehabilitation programs?

Response 2:

 We would like to thank the reviewer for the question. The term “usual rehabilitation programs” refers to “Physical and occupational therapy was initiated at the hospital, focusing on muscle training, standing exercises, walking exercise with orthosis and activities of daily living (ADL) exercises.” As this point was not clear, the word "aforementioned" was added.

Point 3:

Discussion: why only left TA? wasn't intervention done in both ankles? Would you be able to explain this?

Response 3:

It is not clear why muscle activity appeared only in the left TA despite interventions in both lower extremities, and the results of this study do not provide a complete explanation. However, as the HAL-SJ could drive the TA relatively early in the intervention, we speculate that the left lower extremity in this case had a potential for muscle contraction that was masked by the severe motor-sensory deficit. We believe that the advantage of using this device was that visual and motor feedback could be provided to such masked potentials. We have discussed this issue in the revised manuscript as follows:

“We believe that the advantage of using this device is that rich visual and motor feedback was provided to such masked potentials.” (Lines 198–199)

Point 4:

Also, training with the device seems to require intensive sessions with therapists. Time and cost wise, would it this be feasible for other patients? If not, please discuss this as limitations.

Response 4:

As the reviewer indicated, there is a rental fee for this equipment, which may make it difficult to implement in all hospitals. We have added this information to the revised manuscript as follows:

“It should be noted that the device is subject to rental fees, which could be a potential cost and may make it difficult to use in all hospitals and institutions.” (Lines 208–209) 

Reviewer 4 Report

In this study the authors aimed to to assess the feasibility of rehabilitation with a single-joint Hybrid As- 14 sistive Limb (HAL-SJ) ankle joint unit (HAL-T) in a patient with incomplete spinal cord injury (SCI).

Although the study has the potentiality of being shared with the scientific community, I believe that the manuscript would benefit from a major revision with the attempt to better support their experimental setting.

1.     The theoretical framework is scarce, they should clearly describe the scientific evidence that supports the hypothesis they have raised.

2.     Methods section:

-        Experimental procedures should be better defined

-        More information should be provided about the participants’ characteristics.

3.     The Discussion should be enriched with the existing theory. The authors should clearly describe the scientific evidence that supports their findings. In addition, they should start with a first paragraph describing the main aims and then the main results.

4.     I would like to see more of the practical implications. Based on the analyzed variables, how the authors intend to use their findings?

Kind regards

Author Response

Point 1:

 The theoretical framework is scarce, they should clearly describe the scientific evidence that supports the hypothesis they have raised.

Response 1:

The authors would like to thank the reviewer for his/her constructive critique to improve the manuscript. We have made every effort to address the issues raised and to respond to all comments. Please, find next a detailed, point-by-point response to the reviewer's comments. We hope that our revisions will meet the reviewer’s expectations.

We believe that the theoretical background of the effectiveness of the HAL-SJ intervention in this case was large due to rich biofeedback (visual feedback and also sensory feedback that movement actually occurred).

Point 2:

  1. Methods section:

-Experimental procedures should be better defined

-More information should be provided about the participants’ characteristics.

Response 2:

As the reviewer pointed out, the description of the experimental procedure was lacking. In the "Outcome measurement" subsection, we have added the location of the measurement equipment and the name of the software used for the analysis. Moreover, we have provided more information concerning the participant’s age, height, weight, and medical history as well as MRI and radiographic findings to illustrate the participant’s physical condition after the injury. The added part is as follows:

“The study patient was a 34-year-old man (height, 169 cm; weight, 79.4 kg). He only had pre-existing medical history of hyperlipidemia. He sustained a burst fracture of the first lumbar vertebra, dislocation of the right shoulder joint due to a fall from a height. At the time of emergency transport, the patient was found to have bilateral lower extremity paralysis and cysto-rectal disturbance, and was diagnosed as having a spinal cord injury.” (Lines 58–62)

Point 3:

The Discussion should be enriched with the existing theory. The authors should clearly describe the scientific evidence that supports their findings. In addition, they should start with a first paragraph describing the main aims and then the main results.

Response 3:

We would like to thank the reviewer for the comment. As the reviewer indicated, we have added the purpose of this case report and the main results at the beginning of the Discussion section. In addition, the structure of the Discussion has been modified to provide the theoretical background.

Point 4:

I would like to see more of the practical implications. Based on the analyzed variables, how the authors intend to use their findings?

Response 4:

Although the effect obtained with the HAL-SJ alone in this case was partial, it may lead to a step up to another intervention using muscle activity. In addition, improved ankle joint function may contribute to improved gait function in the future. We have discussed this issue in the revised manuscript as follows:

“Although the effects obtained with HAL-SJ alone were partial in this case, they may lead to a step up to another interventions using muscle activity. Thus, this device may be effective as one of the interventions that can be selected based on the patient's condition. In addition, ankle joint function is closely related to standing balance [23], and improvement of ankle joint function can lead to more practical gait by improving balance during walking [24], and may have a spillover effect on ADL, such as allowing selection of a simpler lower limb orthosis in the future. In the future, the number and frequency of effective interventions should be examined. Moreover, it should be further verified the extent to which these interventions are effective.” (Lines 210–218)

Round 2

Reviewer 1 Report

The authors have addressed the concerns raised during the initial review and provided an improved manuscript through the addition of more detail. Thank you for providing me with the information regarding the participant’s age, height, weight, and medical history as well as MRI and radiographic findings to illustrate the participant’s physical condition after the injury which I am interested. My concern still lies in some pictures of the manuscript not being beautiful and scientific enough, thus I would suggest adjusting the following details:

1)     Adjust the size of the two pictures and the annotation text in Figure 1, and increase the scale bar;

2)     Unify the text font format and size of all pictures in the manuscript to improve the overall reading sense of the article;

3)     Split Figures 3a, 3b, and 3c into Figures 4, 5, and 6.

Considering a detailed study has been carried out, I have only the above minor comments addressing paper writing, and once these are addressed the work is recommended for publication.

Best regards!

Author Response

We thank the reviewers for their advice on further improving the manuscript. As you indicated, we have aligned the sizes of the two images in Figure 1 and added scale bars. We have also changed the text font in all figures to the same font used in the text. Figures 3a, b, and c have been split into Figures 3, 4, and 5, respectively.

Reviewer 2 Report

The authors address the comments 

Author Response

Thank you for your peer review. We would like to thank the reviewers whose comments have greatly contributed to the improvement of our manuscript.

Reviewer 3 Report

Authors addressed my comments and concerns. 

Author Response

(The authors gave the same response as above.)

Reviewer 4 Report

Dear Authors,

I believe that the theoretical framework is still insufficient in oder to address your research. Thus I suggest a minor revision of this section.

Kind regards

Author Response

We thank the reviewers for their advice on further improving the manuscript. In the "Introduction," I explained the reason for using HAL-SJ in this case, citing a previous report in which an intervention using HAL-SJ improved the function of paralyzed upper limb muscles in a patient with spinal cord injury. Furthermore, we emphasized that we would like to apply voluntary repetitive motion exercises using HAL-SJ to the ankle joints of the patient with spinal cord injury.

Round 3

Reviewer 1 Report

Very good revision work!